# Impact of Extracorporeal Membrane Oxygenation (ECMO) on Serum Concentrations of Cefepime [note 1]

**DOI:** 10.3390/antibiotics13111024

**Published:** 2024-10-30

**Authors:** Christopher J. Destache, Raul Isern, Dorothy Kenny, Rima El-Herte, Robert Plambeck, Catherine Palmer, Brent S. Inouye, Maura Wong, E. Jeffrey North, Mariaelena Roman Sotelo, Manasa Velagapudi

**Affiliations:** 1School of Pharmacy & Health Professions, Creighton University, Omaha, NE 68178, USA; 2School of Medicine, Creighton University, Omaha, NE 68178, USAmanasavelagapudi@creighton.edu (M.V.)

**Keywords:** cefepime, ECMO, pharmacokinetics

## Abstract

ECMO is becoming widely used as a life-saving measure for critically ill patients. However, there is limited data on pharmacokinetics and the dosing of beta-lactam antibiotics in ECMO. In this study, we evaluated the serum concentrations of cefepime in patients on ECMO to determine the impact of ECMO circuitry and to guide therapeutic dosing. Methods: Patients 19 years or older admitted to the ICU, treated with ECMO and beta-lactam antibiotics for presumed or documented infection, were enrolled. Three blood samples (peak, midpoint, trough) were obtained before ECMO (pre-ECMO) and during ECMO (intra-ECMO) at a steady state. Results: Eight patients met inclusion criteria; six received cefepime. All patients were male. Average ± SD age was 45.8 ± 14.7. Four patients received ECMO for severe SARS-CoV-2 infection, and one each for Pneumocystis pneumonia and influenza A infection. Mean ± SD APACHE II and SOFA scores prior to ECMO were 24.6 ± 7.1 and 11.0 ± 3.9, respectively. All but one of the patients received venovenous (VV) ECMO. Cefepime 1 g every 6 h intravenously over 2 min was administered to all patients before and during ECMO. Cefepime concentrations were fit to non-compartment analysis (NCA) and area under the serum concentration–time curve averaged ± SE 211.9 ± 29.6 pre-ECMO and 329.6 ± 32.3 mg*h/L intra-ECMO, *p* = 0.023. No patients displayed signs of cefepime neurotoxicity. Patients received ECMO for 43.1± 30 days. All patients expired. Cefepime dosed at 1 g every 6 h intravenously appears to achieve therapeutic levels for critically ill patients on ECMO.

## 1. Introduction

Extracorporeal membrane oxygenation (ECMO) is a form of mechanical cardiopulmonary support used in critically ill patients [1]. Components of the ECMO circuit include a venous cannula, a pump, an oxygenator, and an arterial cannula. Venous blood is pumped through an extracorporeal membrane simulating a lung membrane and is returned into the venous or arterial circulation.

Patients undergoing ECMO often present with or are at high risk for severe infections. Optimal dosing of antibiotics is crucial for successful outcomes in patients on ECMO. Though ECMO provides critical support, it is documented to impact pharmacokinetics (PK) of many types of medications, including antibiotics, sedatives, and analgesics. The ECMO circuit acts as an additional PK compartment with the potential to affect the therapeutic levels of antibiotics as they travel through the circuit and back to the patient. The mechanisms by which ECMO changes drug pharmacokinetics include increasing the volume of distribution, sequestering medications, or altering drug clearance. ECMO is also thought to impact drug PK through a direct interaction between the circuit and drug, along with the physiological changes occurring in these patients due to their critical illness [1,2,3,4].

Beta-lactams are commonly utilized in patients on ECMO. They are hydrophilic and primarily undergo renal elimination. Though ECMO more commonly affects PK of lipophilic, highly protein-bound drugs (i.e., ceftriaxone, voriconazole), there are data indicating alterations in PK of beta-lactams [5]. Current data regarding the therapeutic dosing of beta-lactams in adult patients on ECMO are limited; however, ECMO is thought to have a minimal impact on the serum concentrations of cefepime, due to its relative hydrophilicity [3,5]. The findings between different studies are conflicting. One study found inadequate beta-lactam levels in 12% of patients on ECMO, suggesting the potential for increased dosing in early therapy [6]. Another study did not find a difference in beta-lactam levels in ECMO compared to non-ECMO patients [7]. Volume of distribution may be increased because of a systemic inflammatory response (SIRS) often present in critically ill patients with severe infection [8]. According to the current literature, it is unclear if standard dosing of beta-lactams, and particularly cefepime, in patients requiring ECMO is sufficient to achieve therapeutic levels capable of treating susceptible organisms [1,2,3,4,5]. Cefepime is a broad-spectrum fourth-generation cephalosporin (Figure 1).

Its spectrum of activity includes the Enterobacteriaceae family, *Pseudomonas aeruginosa*, and it has Gram-positive activity like first- and second-generation cephalosporins. It has demonstrated a high resistance to pathogen-mediated enzymatic hydrolysis [10]. Clinical uses include soft tissue and skin infections, lower respiratory tract infections, and intra-abdominal and genitourinary infections. Cefepime is widely used in critically ill patients on ECMO. Its bactericidal activity depends on the amount of time the serum concentration is above the minimum inhibitory concentration of the drug, which in turn is based on dosing intervals. Cefepime can be administered as an intermittent infusion over 30 min for all Gram-negative infections except for *Pseudomonas,* for which an extended infusion over 3 to 4 h or continuous infusion over 24 h is preferred [11,12]. Cefepime serum protein-binding values range from approximately 16–19% [1]. Its primary route of elimination is through the kidneys. At high doses, cefepime can cause neurotoxicity. Therefore, therapeutic drug monitoring is crucial, and correct dosing strategies in ECMO are necessary to establish treatment efficacy and avoid toxicity [1,13].

The aim of this prospective study was to identify differences in pharmacokinetic (PK) parameters of beta-lactam antibiotics in adult patients receiving ECMO by comparing PK parameters and serum concentrations of cefepime prior to an ECMO circuit and during the ECMO circuit at a steady state. Assessment of these serum concentrations and PK parameters along with a subsequent correlation of the levels to the time spent above the minimal inhibitory concentration (MIC) will provide guidance on the impact of standard beta-lactam dosing in patients receiving ECMO. It will raise the question of whether dosage adjustments are needed in these patients to prevent either therapeutic failure or drug toxicity.

## 2. Results

A total of eight patients were identified. Six patients received cefepime; two patients received other beta-lactam antibiotics (cefazolin, piperacillin-tazobactam, respectively) and were excluded from the results.

A description of patient demographics and clinical characteristics while undergoing ECMO is presented (Table 1). All patients were male. Average ± SD age was 45.8 ± 14.7. Four patients received ECMO for severe SARS-CoV-2 infection, one for *Pneumocystis jirovecii* pneumonia infection, and one for severe influenza A infection. Two patients received continuous renal replacement therapy (CRRT) during ECMO but after cefepime serum levels were obtained. Mean ± SD APACHE II and SOFA scores prior to ECMO were 24.6 ± 7.1 and 11.0 ± 3.9, respectively (Table 1). Admission serum creatinine averaged 1.2 ± 0.5 mg/dL, and albumin averaged 2.2 ± 0.6 g/dL. ECMO sweep gas flow rate averaged 4.3 ± 0.9, and blood flow rate averaged 4.6 ± 0.7. All but one patient received venovenous (VV) ECMO.

One patient was unable to capture peak cefepime concentration pre-ECMO due to severity of illness and urgency for an ECMO circuit. In Figure 2, an analysis of the serum concentration of cefepime versus time is presented.

Mean (± SD) peak concentration was 82.2 ± 45.7 mcg/mL (pre-ECMO) and 87.6 ± 33.6 mg/L (intra-ECMO) (Figure 2). Average (± SD) trough cefepime concentration was 9.5 ± 0.5 mg/L pre-ECMO and 26.0 ± 27.3 mg/L intra-ECMO.

AUC_0–6_ averaged ± SE 211.9 ± 29.6 pre-ECMO and 329.6 ± 32.3 mg·h/L intra-ECMO, *p* = 0.023. The elimination half-life was 3.85 ± 0.21 h pre-ECMO and 5.84 ± 1.88 h intra-ECMO *p* > 0.05. Cefepime volume of distribution and total clearance averaged 5.1 ± 0.69 pre-ECMO and 4.2 ± 1.0 L intra-ECMO, and 4.68 ± 1.40 pre-ECMO and 3.22 ± 0.38 L/h intra-ECMO, respectively. One patient (intra-ECMO and CRRT) had a trough cefepime concentration > 50 mg/L. Patients received ECMO for 43.1 ± 30 days. All patients expired during hospitalization. Neurotoxicity secondary to cefepime was unable to be assessed due to patients’ clinical conditions at the time of investigation. No other cefepime side effects were noted. Only one patient had a pathogen isolated (*Citrobacter koseri*) from respiratory culture; the remaining patients’ cultures were negative.

## 3. Discussion

The initiation of ECMO introduces several variables which factor into achieving optimal dosing of medications. These include ECMO circuitry, increased volume of distribution, and further variables that affect the pharmacokinetic parameters of antimicrobials.

Previously, three primary hypotheses have been described in which ECMO affects pharmacokinetics: direct sequestration by the circuit, increased volume of distribution, and altered clearance [7]. The ECMO circuitry can vary according to the type of ECMO membrane, tubing, and pump materials. The material used in tubing (polymers, silicones, etc.) can have variable absorption of the medication being administered as well. Studies have demonstrated increased volume of distribution and decreased drug clearance in ECMO [8]. This increased volume of distribution is attributed to system-related factors leading to increased sequestration of the medication being used. Hollow-fiber membranes have demonstrated reduced sequestration. This is one example of how circuit material affects extraction [14]. Drug lipophilicity and protein binding have been demonstrated to affect sequestration of the drug in ECMO circuitry. Altered clearance in ECMO is a variable to consider in antibiotic dosing. Factors such as coexisting or secondary kidney failure, variable organ perfusion, and tissue oxygenation are also contributors to altered clearance in ECMO. Extracorporeal life support organization guidelines report an incidence of acute kidney injury while on ECMO that ranges anywhere from 30% to 85% [13]. ECMO can also potentially affect the clearance of hepatically metabolized drugs, secondary to decreased flow to the liver [7,8,13].

Pharmacokinetic variables are affected by additional patient variables, such as total body weight and creatinine clearance. While we could not identify any side effects or toxicity from cefepime in our study patients, decreased drug clearance can lead to accumulation and can be a basis for a modified dosing regimen in patients with ECMO [15]. The objective of antimicrobial therapy use is to obtain drug levels above the minimal inhibitory concentration at the site of infection. In our study, levels above the MIC were difficult to confirm as only one patient had an organism identified on culture, and all patients eventually expired.

A limitation of this study is the small sample size that was able to be enrolled. Moreover, it was difficult to find patients that could meet the inclusion criteria. While each patient served as their own control, more research on cefepime PK in ECMO patients may be required to make a solid conclusion.

The results obtained in this study allow for the analysis of the serum concentration–time curve, which reflects the amount of antimicrobial (in this case, cefepime) that has effectively reached systemic circulation. Having this result allows for the determination of the area under curve (AUC), a parameter that can be used to evaluate the volume of distribution, total clearance, and bioavailability [16]. The results demonstrate the variability in the pharmacokinetics of cefepime in ECMO. The area under the curve (AUC) reflects the efficacy of the standard dosing of cefepime in ECMO. In Figure 2, an analysis of the serum concentration of cefepime vs. time is presented. A significant increase in the AUC intra-ECMO as compared to pre-ECMO was seen, which could reflect the accumulation with multiple doses or changes in cefepime PK from the ECMO circuit. Cefepime clearance was not significantly changed intra-ECMO compared to pre-ECMO.

## 4. Materials and Methods

### 4.1. Study Population

In this work, the study presented in [17] is expanded upon. This study was performed on patients hospitalized in a one major hospital of a U.S. Midwest healthcare system composed of 12 hospitals located in 2 states. The Institutional IRB (Institutional Research Board for Human Studies) granted approval and determined this study as human subject research. After IRB approval, we identified patients based on our inclusion and exclusion criteria. The first patient was recruited on 30 September 2021, and the last patient was recruited on 28 August 2023. A signed informed consent form was obtained from patients or their representative(s). The inclusion criteria for enrollment were age ≥ 19 years receiving ECMO and receiving a beta-lactam antibiotic. Exclusion criteria included age < 19 years, pregnancy, identifying as a Jehovah’s Witness, hospice or comfort care, and receiving a beta-lactam antibiotic for more than 72 h.

The Getinge Cardiohelp^®^ ECMO (Maquet Getinge Cardiopulmonary AG, Rastatt, Germany) device was used for all patients according to the manufacturer’s recommendations. ECMO tubing used was sterile heparin-bonded polyvinyl chloride (PVC). Cefepime was administered to patients as a 1 g dose over 2 min every 6 h as part of hospital policy.

### 4.2. Data Collection and Analysis

Data collected from the electronic medical record included age, gender, number of days on ECMO, serum creatinine on admission, admission SOFA score and APACHE II score, serum albumin, ECMO parameters, and mortality. Blood (4 mL) was collected during two periods; before the placement of ECMO (pre-ECMO) and after placement of the ECMO circuit (intra-ECMO). Samples were collected using red-top vacutainer (B-D, Franklin Lakes, NJ, USA) tubes. Tubes were allowed to clot at room temperature for 10–15 min, and they were centrifuged at 3000× *g* and stored at −80°C until LC-MS/MS analysis. Cefepime serum concentrations (peak 0.5 h, midpoint 3–4 h, and trough 5.5 h after dosing) were obtained after patients received at least 24 h of continuous cefepime dosing. Cefepime concentrations were determined at Creighton University using the ABI QTRAP^®^ 5500 LC-MS/MS system coupled with an Exion liquid chromatography station, including an LC controller, degasser, two liquid pumps, column oven, LC autosampler, and a peak gas generator running MultiQuant 3.0.2 software. Cefepime assay methodology has been reported [11]. The lower limit of detection for the assay was 0.01 µg/mL. Intra-day and inter-day variability of the cefepime assay was < 10%. Mean (± SD) or percentages are reported.

GraphPad Prism was used for fitting using non-compartmental analysis (NCA) and weighting by 1/Y^2^ (Table 2). Pharmacokinetic parameters including area under the serum concentration–time curve were derived from NCA. Area under the serum concentration-time curve pre-ECMO was calculated in GraphPad using extrapolated D/AUC_0–∞_ and intra-ECMO as AUC_0–6_ and compared between pre-ECMO and intra-ECMO. The ratio of Cl_intra-ECMO_/Cl_pre-ECMO_ demonstrated that four of the five patients had lower Cl intra-ECMO, with one patient’s clearance being greater during pre-ECMO and one patient’s pre-ECMO Cl being unable to be determined.

## 5. Conclusions

The current conventional dosing of cefepime at 1 g every 6 h is shown to achieve adequate serum levels in critically ill patients on ECMO, suggesting that dose adjustments are not required. Variables affecting pharmacokinetics in ECMO should be considered on a case-by-case basis.

## Figures and Tables

**Figure 1 antibiotics-13-01024-f001:**
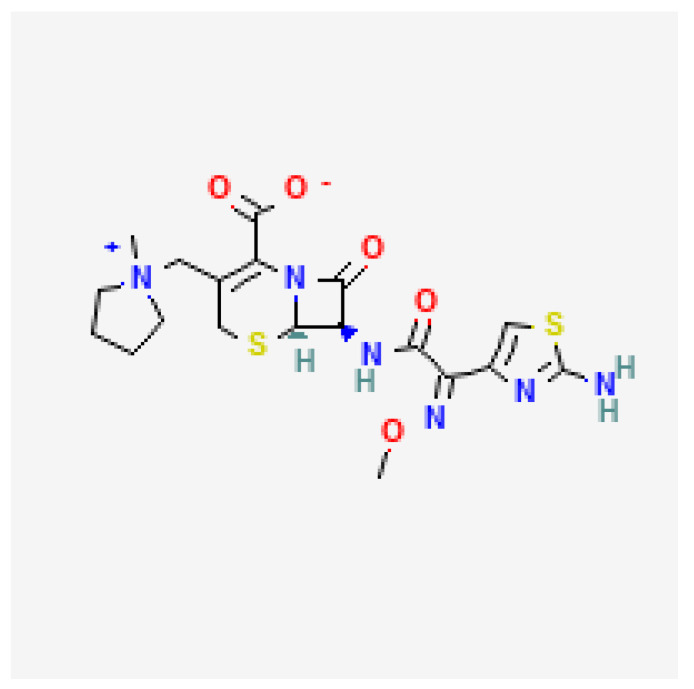
Cefepime structure [9].

**Figure 2 antibiotics-13-01024-f002:**
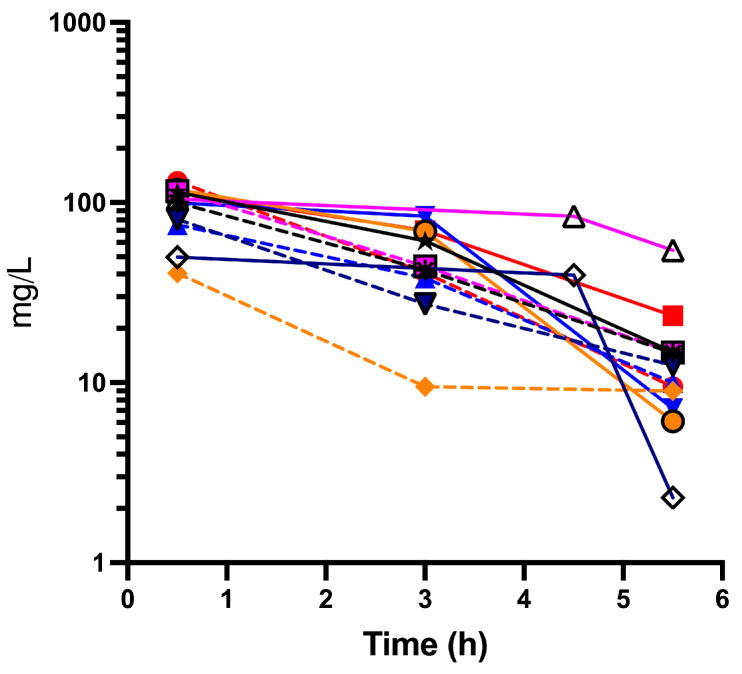
Cefepime biodistribution pre-ECMO (dashed lines) and intra-ECMO (solid lines). Each color represents results from each patient.

**Table 1 antibiotics-13-01024-t001:** Baseline clinical characteristics of patients and ECMO variables.

Demographic Data	Median	Patient 1	Patient 2	Patient 3	Patient 4	Patient 5	Patient 6
Sex		M	M	M	M	M	M
Age (yrs)	43	35	43	35	52	32	59
SOFA Score	10	12	10	8	8	7	16
APACHE II Score	22	17	20	22	22	22	41
Serum Albumin (g/dL)	2.1	2.5	2.4	1.9	2.0	1.5	2.1
Days on ECMO	46	69	70	84	46	62	9
Admission creatinine clearance (mL/min)	>90	>90	>90	>90	>90	88	47
ECMO flow rate (L/min)	4.2	4.2	4.2	4.2	4.7	3.9	3.8
ECMO Mode	VV	VV	VV	VV	VV	VV	VA

VV = venovenous; VA = venoarterial.

**Table 2 antibiotics-13-01024-t002:** Cefepime PK comparison pre-ECMO and intra-ECMO.

PK Parameter		Patient 1	Patient 2	Patient 3	Patient 4	Patient 5	Patient 6	*p*-Value
AUC_0–6_ (mg·h/L)	Pre-ECMO	277.4	201.6	85.6	273.6	184.6	248.8	0.023
	Intra-ECMO	346.1	343.5	328.0	446.0	199.3	314.4	
Vd (L)	Pre-ECMO	7.6	5.0		3.7	5.4	4.0	>0.05
	Intra-ECMO	8.8	2.9	3.0	2.2	5.0	3.2	
Cl (L/h)	Pre-ECMO	3.13	3.66		2.81	3.82	2.98	>0.05
	Intra-ECMO	2.25	2.67	2.84	0.62	4.60	2.51	
T_1/2_ (h)	Pre-ECMO	3.04	3.96		3.87	4.28	4.14	>0.05
	Intra-ECMO	5.06	3.04	2.69	15.0	5.40	3.90	
Cl (intra-ECMO)/Cl (pre-ECMO)		0.72	0.73		0.22	1.20	0.84	

AUC = area under the serum concentration–time curve 0–6 h; Vd = volume of distribution; Cl = total body clearance; T_1/2_ = elimination half-life.

## Data Availability

Data availability is available upon request from the corresponding author.

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
