# Peer review of "Impact of Extracorporeal Membrane Oxygenation (ECMO) on Serum Concentrations of Cefepimeâ€"

_antibiotics, 2024, doi:10.3390/antibiotics13111024_

Round 1
Reviewer 1 Report
Comments and Suggestions for Authors
This manuscript evaluated cefepime PK in patients pre-ECMO and during. It was found that volume of distribution was increased and clearance remained similar after ECMO, and concluded that the standard dosing can achieve therapeutic levels of cefepime in these critically ill patients on ECMO. To have a solid conclusion from this study, authors are advised to address the following major comments:
1. When collecting intra-ECMO PK samples, how many doses of cefepime have been given and how did you know that the steady state has been reached? Please specify.
2. Please provide CL, Vd, and AUC values for each of the 6 subjects either in Table 1 or in a separate table, and use a comparison metric such as CL(intra-ECMO)/CL(pre-ECMO) for each individual so we can see a paired comparison (as each patient serves as his own control group).
3. In Figure 1, please also provide a plot with a log-scaled y-axis, which can reveal if the one-compartment PK model is suitable (if so, only one exponential decline phase is expected). Also, if possible, please use different colors (or symbols) for different individuals, so when looking at the plots we know how each individual changed with ECMO.
4. In Materials and Methods, please elaborate on the methodology of one-compartment model fitting and general PK analysis. (1) What software was used for model fitting; (2) For intra-ECMO PK, how did you account for the drug accumulation from previous dosing (pre-ECMO and multiple dosing intra-ECMO etc.); (3) Were AUC values derived based on the one-compartment model or non-compartmental analysis?
5. The last paragraph in Discussion needs to clarify the following questions: (1) Line 148: AUC can be used to evaluate Vd, CL, and F. However, this AUC should be AUC from time 0 to infinity, not AUC0-6h as presented in the manuscript for single dose or first dose. In noncompartmental analysis, pre-ECMO CL can be calculated as Dose/AUC0-infinity, whereas for intra-ECMO if steady state has been reached, CL should be calculated as Dose/AUC0-tau. (2) Line 153: increased AUC intra-ECMO compared to pre-ECMO reflects the variability in cefepime concentrations in ECMO. Not really, the AUC was increased perhaps simply because of multiple dosing and drug accumulation, nothing related to ECMO itself. (3) Line 150: “The results demonstrate the variability in pharmacokinetics of cefepime in ECMO”. It is unclear what authors mean by “variability” here. “Variability” usually indicate between-subject variabilities, but has been used in an ambiguous manner throughout. Please be clear.
6. Page 2 Line 49-50 reads confusing. Suggest to rewrite this whole paragraph for better flows and message delivery.
7. Page 2 Line 50-58: it seems that literature gives conflicting findings about dosing in patients on ECMO, and the manuscript aims to understand if standard dosing is sufficient. But considering the very small sample size in this study, the conclusion is still not solid. Please address/stress this limitation.
8. Time of concentrations above MIC was not investigated in the study. Page 4 Line 143 does touch upon this. How should MIC be determined usually? Why in Conclusions it can be concluded that the current dosing regimen is “adequate”?
Author Response
- When collecting intra-ECMO PK samples, how many doses of cefepime have been given and how did you know that the steady state has been reached? Please specify. All patients had serum samples obtained after a minimum of 3 doses of cefepime.
- Please provide CL, Vd, and AUC values for each of the 6 subjects either in Table 1 or in a separate table, and use a comparison metric such as CL(intra-ECMO)/CL(pre-ECMO) for each individual so we can see a paired comparison (as each patient serves as his own control group). The authors have added this as Table 2.
- In Figure 1, please also provide a plot with a log-scaled y-axis, which can reveal if the one-compartment PK model is suitable (if so, only one exponential decline phase is expected). Also, if possible, please use different colors (or symbols) for different individuals, so when looking at the plots we know how each individual changed with ECMO. The authors have made changes in Figure 2 to comply with these suggestions.
- In Materials and Methods, please elaborate on the methodology of one-compartment model fitting and general PK analysis. (1) What software was used for model fitting; (2) For intra-ECMO PK, how did you account for the drug accumulation from previous dosing (pre-ECMO and multiple dosing intra-ECMO etc.); (3) Were AUC values derived based on the one-compartment model or non-compartmental analysis? This was added to the Methods section for clarity.
- The last paragraph in Discussion needs to clarify the following questions: (1) Line 148: AUC can be used to evaluate Vd, CL, and F. However, this AUC should be AUC from time 0 to infinity, not AUC0-6h as presented in the manuscript for single dose or first dose. In noncompartmental analysis, pre-ECMO CL can be calculated as Dose/AUC0-infinity, whereas for intra-ECMO if steady state has been reached, CL should be calculated as Dose/AUC0-tau. (2) Line 153: increased AUC intra-ECMO compared to pre-ECMO reflects the variability in cefepime concentrations in ECMO. Not really, the AUC was increased perhaps simply because of multiple dosing and drug accumulation, nothing related to ECMO itself. (3) Line 150: “The results demonstrate the variability in pharmacokinetics of cefepime in ECMO”. It is unclear what authors mean by “variability” here. “Variability” usually indicate between-subject variabilities, but has been used in an ambiguous manner throughout. Please be clear. The authors have made changes in the Discussion section to improve the clarity of our results.
- Page 2 Line 49-50 reads confusing. Suggest to rewrite this whole paragraph for better flows and message delivery. This has been re-written.
- Page 2 Line 50-58: it seems that literature gives conflicting findings about dosing in patients on ECMO, and the manuscript aims to understand if standard dosing is sufficient. But considering the very small sample size in this study, the conclusion is still not solid. Please address/stress this limitation. The authors have added a limitations section to the Discussion.
- Time of concentrations above MIC was not investigated in the study. Page 4 Line 143 does touch upon this. How should MIC be determined usually? Why in Conclusions it can be concluded that the current dosing regimen is “adequate” The MIC would have been reported by the Microbiology Laboratory from the automated susceptibilities. However, there were only 1 pathogen isolated. THe authors believe the current regimen is adequate as all the trough serum levels were >2 mg/L in the PK analysis.
Reviewer 2 Report
Comments and Suggestions for Authors
It is an article with a relevant topic about the use of medical devices (ECMO) and how they can affect the pharmacokinetics of cefepime. There are only certain points that have to be considered:
1) Important to mention some examples of protein bound drugs affected by ECMO.
2) It would be convenient to include the chemical structure of cefepime.
3) What were the values ​​of the area under the curve from time zero to infinite time and the values ​​of the elimination constant?
4) Visually, it would be very convenient to present the results obtained in a table.
Author Response
1) Important to mention some examples of protein bound drugs affected by ECMO. Several drugs were added to the introduction affected by ECMO.
2) It would be convenient to include the chemical structure of cefepime. This has been added as Figure 1.
3) What were the values ​​of the area under the curve from time zero to infinite time and the values ​​of the elimination constant? The authors have added another table for cefepime PK results.
4) Visually, it would be very convenient to present the results obtained in a table. This has been added.
Round 2
Reviewer 1 Report
Comments and Suggestions for Authors
1. Some results have been updated and numbers reported are inconsistent with the Abstract. Please go through the manuscript and make corrections as needed.
2. In Table 2, Cl(intra-ECMO)/Cl(pre-ECMO) seemed to be calculated as pre-ECMO/intra-ECMO instead. Please correct.
3. Page 6 Line 177: authors claim that cefepime volume of distribution modestly increased. This is not true. In Table 2, actually most of patients exhibit a decrease in Vd. Mean values increased in intra-ECMO because 1 patient showed an increase and 1 patient did not have a result pre-ECMO. Please revisit this conclusion.
4. Page 5 Line 122: “Table 2” was referred to when reporting trough cefepime concentration, but there is no Ctrough in Table 2.
5. The following comments were not adequately addressed, and I am having a major concern/confusion as to whether non-compartmental analysis (NCA) was used or one-compartment modeling was used: “For intra-ECMO PK, how did you account for the drug accumulation from previous dosing (pre-ECMO and multiple dosing intra-ECMO etc in the one-compartment analysis”. Please show me the equations used in GraphPad Prism for the PK analysis to clarify. If you did not model the actual multiple doses in ordinary differential equations, e.g., just using CL=Dose/AUC, it is NCA rather than a 1-compartment model.
6. Also, as I mentioned in my previous comments, 1st dose clearance should be calculated as Dose/AUC0-infinity (with AUC6h-infinity being extrapolated), whereas for intra-ECMO steady state, CL should be calculated as Dose/AUC0-tau. If NCA was used, please make sure calculation was conducted this way, and this should be explicitly defined in Methods as well.
Author Response
- Some results have been updated and numbers reported are inconsistent with the Abstract. Please go through the manuscript and make corrections as needed. The authors have gone through the manuscript and made corrections throughout so that all numbers are updated.
- In Table 2, Cl(intra-ECMO)/Cl(pre-ECMO) seemed to be calculated as pre-ECMO/intra-ECMO instead. Please correct. The authors have made changes in Table 2 to ensure these are correct. we regret this error.
- Page 6 Line 177: authors claim that cefepime volume of distribution modestly increased. This is not true. In Table 2, actually most of patients exhibit a decrease in Vd. Mean values increased in intra-ECMO because 1 patient showed an increase and 1 patient did not have a result pre-ECMO. Please revisit this conclusion. The authors have made changes to this line in the revised manuscript.
- Page 5 Line 122: “Table 2” was referred to when reporting trough cefepime concentration, but there is no Ctrough in Table 2. We deleted this Table 2 designation. Apologies.
- The following comments were not adequately addressed, and I am having a major concern/confusion as to whether non-compartmental analysis (NCA) was used or one-compartment modeling was used: “For intra-ECMO PK, how did you account for the drug accumulation from previous dosing (pre-ECMO and multiple dosing intra-ECMO etc in the one-compartment analysis”. Please show me the equations used in GraphPad Prism for the PK analysis to clarify. If you did not model the actual multiple doses in ordinary differential equations, e.g., just using CL=Dose/AUC, it is NCA rather than a 1-compartment model. The authors have made changes to these parameters using NCA. We have made changes to the methods section to describe what was accomplished.
- Also, as I mentioned in my previous comments, 1st dose clearance should be calculated as Dose/AUC0-infinity (with AUC6h-infinity being extrapolated), whereas for intra-ECMO steady state, CL should be calculated as Dose/AUC0-tau. If NCA was used, please make sure calculation was conducted this way, and this should be explicitly defined in Methods as well. The authors have made these necessary changes in the Results and Methods sections.
Round 3
Reviewer 1 Report
Comments and Suggestions for Authors
Please delete the discussion on Vd pre- and intra-ECMO (Page 6 Line 176) - it is not conclusive whether Vd increased or decreased. Just say Vd was not significantly changed as p>0.05.
Please correct Page 7 Line 214: "one-compartment model" is still there.
Author Response
Please delete the discussion on Vd pre- and intra-ECMO (Page 6 Line 176) - it is not conclusive whether Vd increased or decreased. Just say Vd was not significantly changed as p>0.05. This has been completed
Please correct Page 7 Line 214: "one-compartment model" is still there. This also has been completed.